# Point sources of ultra-high-energy neutrinos: minimalist predictions for near-future discovery

**Damiano F. G. Fiorillo⋆, Mauricio Bustamante and Victor B. Valera**

Niels Bohr International Academy, Niels Bohr Institute, University of Copenhagen, 2100 Copenhagen, Denmark

⋆ damiano.fiorillo@nbi.ku.dk

## Abstract

**The discovery of ultra-high-energy neutrinos, with energies above 100 PeV, may soon be within reach of upcoming neutrino telescopes. We present a robust framework to compute the statistical significance of point-source discovery via the detection of neutrino multiplets. We apply it to the radio array component of IceCube-Gen2. To identify a source with $3\sigma$ significance, IceCube-Gen2 will need to detect a triplet, at best, and an octuplet, at worst, depending on whether the source is steady-state or transient, and on its position in the sky. The discovery, or absence, of sources significantly constrains the properties of the source population.**

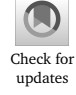

## 1 Introduction

The origin of ultra-high-energy cosmic rays (UHECRs) is a fundamental unknown of theoretical physics. They must be accelerated to energies higher than $10^{12}$ GeV, and the acceleration likely happens within extragalactic astrophysical sources. While several candidates have been proposed in the literature, none of them has been irrefutably identified. A crucial difficulty is that UHECRs are deflected in cosmic magnetic fields, and do not track back to their sources. This difficulty could be overcome by the recent field of multimessenger astronomy, and especially via the detection of astrophysical neutrinos. These are expected to be produced in the interaction of UHECRs with dust and radiation in the sources in which they are accelerated, and subsequently reach the Earth. Differently from UHECRs, neutrinos are electrically neutral, and thus they point back to their sources, allowing in principle to pinpoint the sources they come from.

A first step in this direction was the discovery of a diffuse flux of neutrinos of TeV-PeV energies, made by the IceCube neutrino telescope. IceCube has identified candidate point sources for a few of these neutrinos, including the active galactic nucleus TXS 0506+056, the starburst galaxy NGC 1068, and the tidal disruption event AT 2017dsg. Since neutrinos typically carry an energy equal to 5% of their parent proton energy, these sources are guaranteed accelerators of UHECRs up to energies with tens of PeV. However, there is no certainty that these sources are also the long-sought sources of EeV-scale UHECRs. The definitive answer to this question can be obtained by detecting point sources of ultra-high-energy (UHE) neutrinos, above 100 PeV.

Albeit proposed in the 1960s, UHE neutrinos remain undiscovered. The main challenge is the steep decrease of the astrophysical neutrino flux, so that UHE neutrinos are rare and difficult to detect. However, in the coming decade a number of experiments are expected to start taking data, and will be sensitive enough to attain the detection of UHE neutrinos within the next 10-20 years. This will provide in principle the possibility of performing UHE neutrino astronomy via the detection of neutrino multiplets, i.e. clusters of neutrinos from similar positions in the sky. In Ref. [1], we provide state-of-the-art forecasts for this possibility. As a benchmark experiment, we focus on the radio array at IceCube-Gen2, which promises to be one of the most advanced and most sensitive ones, expected to start operations in the 2030s. However, our methods are of wider applicability and can be used in the more general context of UHE neutrino telescopes.

The possibility of discovering cosmic accelerators by using the angular distribution of UHE neutrinos has first been pointed out in Ref. [2], which used the angular distribution of UHE neutrinos as a probe of the presence of point sources. However, due to the lack of realistic estimates of the experiment response at the time, this paper adopted some simplifying assumptions on the experiment performance (e.g., a uniform detector effective area with a fixed fractional sky coverage). We provide the first prospects for point-source discovery, accounting for the angular-dependent response of the experiment and the crucial role of neutrino propagation through the Earth. We phrase our work in terms of two main questions: first, how large should a neutrino multiplet be in order to conclusively claim a point-source identification? Second, what information can we draw on UHE neutrino source populations if a source is detected or if no source is detected?

## 2 Prospects for UHE neutrino detection

In order to determine how large must a neutrino multiplet be to claim a point-source detection, we need an accurate modeling both of the detector response to a given neutrino flux, and of the background neutrino fluxes which may produce fictitious multiplets unrelated to astrophysical point sources. We discuss both these aspects in this section.

### 2.1 Detector response

A key improvement of our work over the preceding literature is to provide a detailed description of the response of the radio array at IceCube-Gen2 to an astrophysical neutrino flux. This includes accounting for neutrino propagation through the Earth, and a full simulation of the detector response. Here we outline our approach.

When astrophysical neutrinos arrive at the Earth, they reach the detector, located at the South Pole, from different directions. Neutrinos passing through the Earth are attenuated and shifted to lower energies because of the scattering with nuclei in matter. This induces a peculiar anisotropy in the response of the detector. Downgoing neutrinos arrive at the detector from above, and are largely unattenuated. Upgoing neutrinos arrive at the detector from below, and are strongly attenuated. Earth-skimming neutrinos arrive at the detector from directions

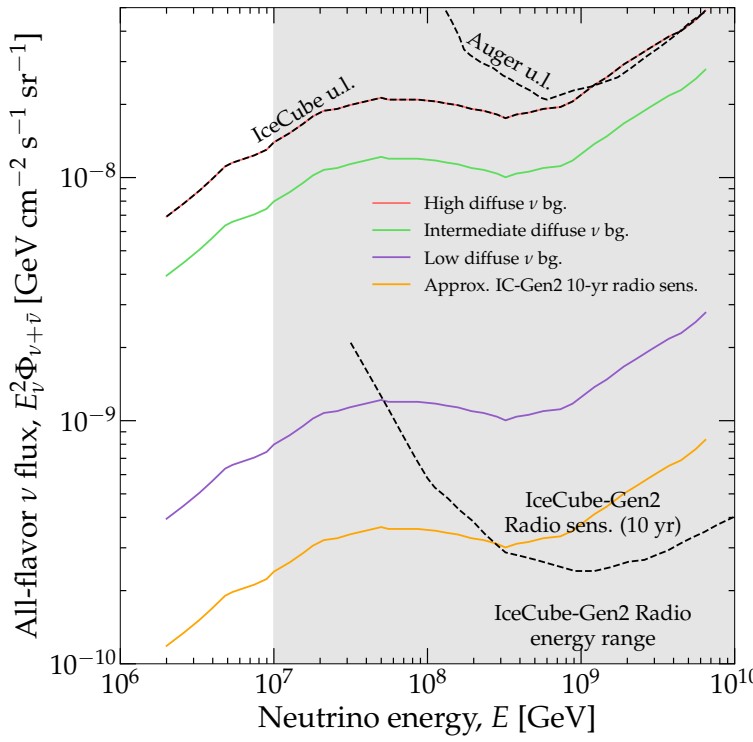

Figure 1: Benchmark diffuse UHE neutrino flux models used in our work. For comparison, we include the IceCube-Gen2 radio array sensitivity [5], and the upper limit from Auger [6].

close to the horizon, so that they are mildly attenuated. We account for neutrino in-Earth propagation using the state-of-the-art tool NUPROPEARTH [3].

Inside the detector, a neutrino interacts with a proton or neutron of the Antarctic ice, via $\nu N$ DIS, triggering a high-energy particle shower that receives a fraction of the neutrino energy. The charged particles in the shower emit a coherent radio signal—Askaryan radiation, which is then detected by the radio array. A complete modeling of the detector must account both for the geometry of the detector, and for the neutrino-nucleon scattering. For this work, we adopt the IceCube-Gen2 effective volumes from Ref. [4], computed via simulations using the same tools as the IceCube-Gen2 Collaboration.

## 2.2 Backgrounds

The main challenge to multiplet searches is that, underlying the UHE neutrinos from point sources, we expect a diffuse background of UHE neutrinos and atmospheric muons whose random over-fluctuations may mimic multiplets from point sources.

The diffuse flux of UHE neutrinos is likely composed of cosmogenic neutrinos, made in UHECR interactions en route to Earth, and of neutrinos from unresolved sources. While this flux has never been measured, upper bounds have been obtained on it mainly from IceCube [7] and Auger [6]. Rather than adopting a particular prediction, we set the diffuse UHE neutrino flux to benchmark levels representative of current and future detector sensitivity: the current IceCube upper limit on the energy flux $E_\nu^2 \Phi_\nu$ (*high*) [7], and versions of it shifted down to $10^{-8}$ (*intermediate*) and $10^{-9}$ GeV cm$^{-2}$ s$^{-1}$ sr$^{-1}$ (*low*). We show the three versions of the background in Fig. 1. We also account for a background of atmospheric muons, which is, however, negligible.

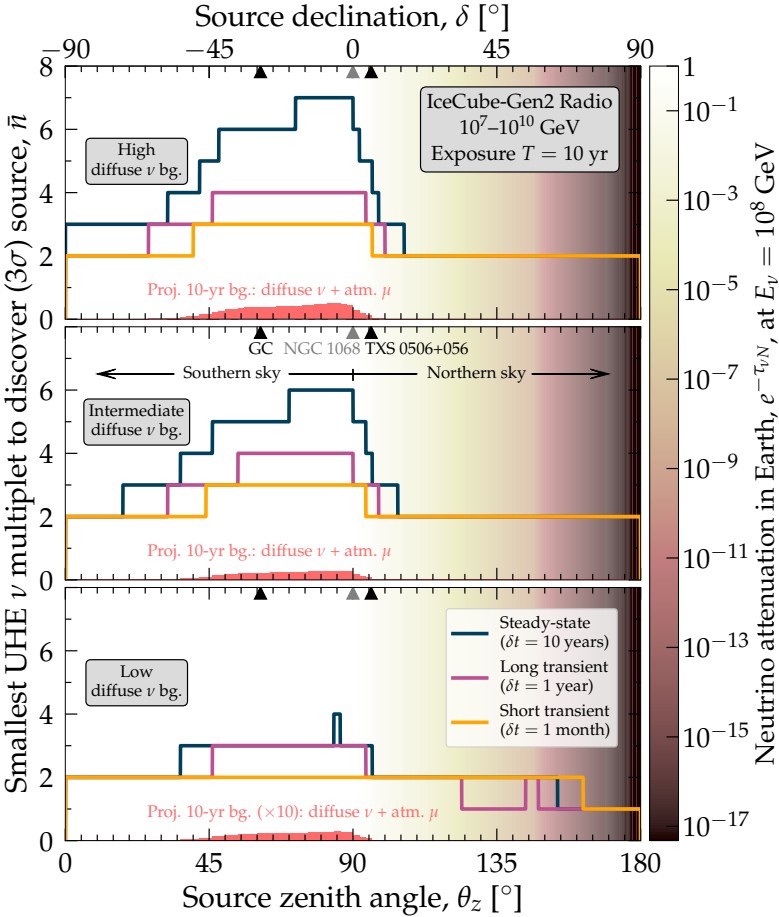

Figure 2: Smallest UHE multiplet needed for the IceCube-Gen2 radio array to discover an UHE neutrino source, steady-state or transient, with global significance of $3\sigma$, for three choices of the unknown background diffuse UHE neutrino flux: high (*top*), intermediate (*center*), and low (*bottom*). For each, we show the projected 10-year rate of background events with reconstructed shower energy of $10^7$–$10^{10}$ GeV. The shading shows the in-Earth attenuation coefficient for 100-PeV neutrinos, where $\tau_{\nu N}$ is the neutrino optical depth.

### 2.3 Discovering sources

We can now answer our first question, namely the minimum multiplet size needed to claim a discovery. We tessellate the sky in pixels, with a size equal to the angular resolution of the detector, namely $\sigma_{\theta_z} = 2°$ (see, e.g., Ref. [8]). In the $i$-th pixel, we compute the expected number of events due to the background $\mu_i$ after the exposure time $T$. To exclude the possibility that a neutrino multiplet was background-induced, rather than source-induced, we need the probability that the background alone produced a multiplet.

The *local* p-value $p$ of detecting a multiplet of more than $n_i$ events in the $i$-th pixel, i.e., the probability that a multiplet is due to background alone, is $p(\mu_i, n_i) = \sum_{k=n_i}^{+\infty} (\mu_i^k/k!)e^{-\mu_i}$. But this does not account for the look-elsewhere effect: even if $p$ is small—so that a background fluctuation is unlikely—the probability that an excess with this p-value occurs anywhere in the sky may be large. Therefore, following Ref. [1], in our forecasts we use instead the *global* p-value $P(p)$, i.e., the probability that a multiplet with local p-value $p$ occurs in any of the pixels.

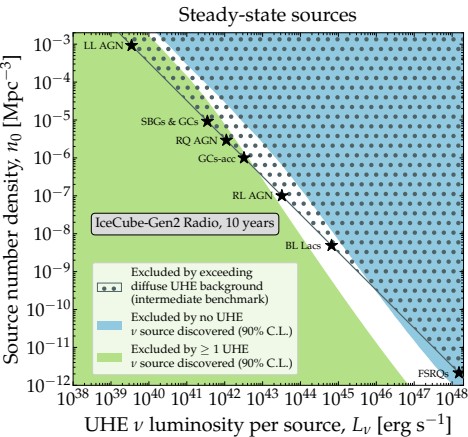
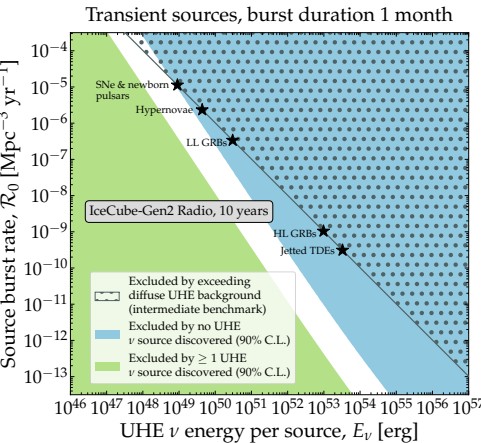

Figure 3: Constraints on candidate classes of steady-state (*left*) and transient (*right*) UHE neutrino sources, from the discovery or absence of UHE neutrino sources. We showcase promising candidate source classes. For each, the value of $n_0$ or $\mathcal{R}_0$ is from Refs. [9,10]; the value of $L_\nu$ or $E_\nu$ is chosen to saturate the background UHE neutrino diffuse flux; for this plot, we fix it to our intermediate benchmark. In the hatched region, the flux from the source population exceeds the background diffuse flux.

Figure 2 shows the smallest multiplet needed to claim source discovery at $3\sigma$, i.e., with $P = 0.003$, in $T = 10$ yr of exposure time, for our three background neutrino diffuse fluxes. In all cases, sources located above IceCube-Gen2 ($\theta_z \lesssim 45°$), where the background is smallest, may be discovered by detecting a doublet or triplet, regardless of the choice of benchmark. However, detection is unlikely in these directions because the detector effective volume is small. In contrast, sources located closer to the horizon ($45° \lesssim \theta_z \lesssim 95°$), where the background is largest, require larger multiplets, as large as a heptaplet for the high background benchmark. Yet, detection is promising in these directions because the effective volume is larger and in-Earth attenuation is mild.

## 3 Constraints on source population

The detection of point sources implies that the neutrino sky is populated with bright neutrino emitters. On the contrary, if no point source is observed, an upper bound can be set on the neutrino luminosity of individual neutrino sources. Here we quantify these statements. A population of steady sources can be described in terms of its local number density $n_0$ and the individual neutrino luminosity $L_\nu$. For transient sources, one can use the local burst rate $\mathcal{R}_0$ and the energy emitted in neutrinos in a burst $E_\nu$. The entire source population produces a diffuse neutrino flux proportional to the product $n_0 L_\nu$, or $\mathcal{R}_0 E_\nu$. Such a diffuse flux cannot exceed the assumed background models in Fig. 1. Therefore, for each background model, a first constraint can be drawn by this requirement.

We consider the scenario in which UHE neutrinos are produced by this population of sources, candidate for detection, and by an unresolved background of neutrinos. The latter can mimic either a cosmogenic neutrino flux or a flux from dim astrophysical sources, and is determined so that the total diffuse neutrino background equals the background flux assumed in each of the three models of Fig. 1. For each background model, we can now determine the probability of observing in at least one pixel in the sky a multiplet larger than the threshold value of Fig. 2 in an exposure time of 10 years. We use an exact analytical approach detailed

in Ref. [1] for computing this probability as a function of $n_0$ and $L_\nu$ for steady sources, and of $\mathcal{R}_0$ and $E_\nu$ for transient sources. If at least one source is discovered after 10 years, we require this probability to be larger than 10%; if no source is discovered, it should be larger than 90%.

Fig. 3 shows the corresponding 90% confidence level exclusions. For steady sources, most candidates lie in the green region; they are not expected to be observed, and even a single source detection would exclude them as dominant candidates to the UHE neutrino flux. For transient sources, most of the sources lie in the blue region; because of their shorter duration, they are expected to be observed, and if none of them is detected they cannot be the dominant component of the UHE neutrino production. We assume a benchmark luminosity common to all sources. Accounting for a luminosity distribution would likely increase the chance of particularly bright outlier sources, which would improve the discovery potential.

## 4 Conclusion

We provide the first state-of-the-art prospects for detection of point sources of UHE neutrinos. While we focus on detection at IceCube-Gen2, our methods can be easily extended to other telescopes in the same energy range. Our prospects are obtained by a detailed simulation of the experiment. Furthermore, the forecasts are extended to obtain projected constraints on the properties of the dominant population of UHE neutrino sources. Detecting even a single source would exclude most steady source candidates for neutrino production, whereas detecting no source would exclude most transient sources as dominant contributors to the UHE neutrino sky.

## Acknowledgements

**Funding information** The authors are supported by the Villum Fonden under project no. 29388. This project has received funding from the European Union's Horizon 2020 research and innovation program under the Marie Sklodowska-Curie grant agreement No. 847523 'INTERACTIONS'. This work used resources provided by the High Performance Computing Center at the University of Copenhagen.

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
