# Peer review of "Point sources of ultra-high-energy neutrinos: minimalist predictions for near-future discovery"

_SciPost Physics Proceedings, doi:SciPost Phys. Proc. 13, 009 (2023)_

## Round 1 · Referee Report · Anonymous · 2022-8-30

Strengths

1) presentation of material + clarity of article
2) choice of robust methods (binned counting experiment)
3) improvement over past work on this topic (more realistic detector assumptions)
4) technical details explained elsewhere - referenced

Weaknesses

1) assumption of standard candle luminosity function in the context of expected constraints on source populations

Report

This review concerns the conference proceeding titled “Point sources of ultra-high-energy neutrinos: minimalist predictions for near-future discovery” that supplements a corresponding contribution to ISVHECRI 2022.

The authors analyzed the potential of future neutrino detectors for detecting and/or constraining the population of ultra-high-energy neutrino sources (above ~100 PeV). In particular they focus on the proposed IceCube-Gen2 radio component to demonstrate their methods.

The proceeding is well written and the results are important as the authors improved over previous work on this topic, for example by using more realistic detector response functions.
A lot of the technical details that enable this work are presented elsewhere (ref 2, arXiv:2205.15985). This works, because enough pointers are given so readers can follow the big picture.

The two main results, the 3 sigma discovery potential in units of number of signal neutrinos (Fig. 2) and the relation between constraints on the underlying population of sources depending on whether or not sources are discovered (Fig. 3), rely on binning the sky, counting neutrinos and applying poisson statistics - all of which seem appropriate choices.

I support publication in Scipost Physics Proceedings but would advise one additional clarification regarding the interpretation of Fig 3. Otherwise I have mostly minor comments that will improve readability.

Comment on Fig 3 + interpretation:
These results assume the sources to behave as standard candles, yet many realistic source populations are described by luminosity functions that differ significantly from a delta function (for example powerlaws). I assume that accounting for these distributions will add extra variance to the problem and widen underlying distributions. Did the authors study whether some conclusions, e.g. “Detecting even a single source would exclude most steady source candidates for neutrino production” are robust against the choice of luminosity function (here delta / standard candle)? What about a “more luminous than average”-source being responsible for the detection?
The authors may choose to add a sentence to Section 3 or 4.

Minor comments:

Second to last paragraph of Sec 1: “IceCube-Gen2, which is one of the most advanced” -> IceCube-Gen2, promises to be one of the most advanced (or similar)

Fig 1: Suggest to add a black dashed line on top of the red line to make clear that the red line is the experimental UL (consistent style with Auger, Gen2 lines). It becomes more obvious that colored lines are benchmark scales.

Sec 2.3 2nd sentence: The quoted angular resolution of 2deg would benefit from a reference.
I am not aware of any Gen2 reference but similar designs could be used in support, for example:
RNO-G: pos.sissa.it/395/1026 (ICRC 2021)
ARIANNA: pos.sissa.it/395/1151 (ICRC 2021)

Sec 2.3 2nd paragraph: suggest to add reference that explains how local pvalues are converted to global ones. Should be in Ref 2.

Sec 1:
Track back their sources -> track back to their sources
And especially from the detection of -> especially through (or via) the detection of

Requested changes

1) Comment on Fig 3 + interpretation:
These results assume the sources to behave as standard candles, yet many realistic source populations are described by luminosity functions that differ significantly from a delta function (for example powerlaws). I assume that accounting for these distributions will add extra variance to the problem and widen underlying distributions. Did the authors study whether some conclusions, e.g. “Detecting even a single source would exclude most steady source candidates for neutrino production” are robust against the choice of luminosity function (here delta / standard candle)? What about a “more luminous than average”-source being responsible for the detection?
The authors may choose to add a sentence to Section 3 or 4.

2) Sec 2.3 2nd sentence: The quoted angular resolution of 2deg would benefit from a reference.
I am not aware of any Gen2 reference but similar designs could be used in support, for example:
RNO-G: pos.sissa.it/395/1026 (ICRC 2021)
ARIANNA: pos.sissa.it/395/1151 (ICRC 2021)

  • validity: high
  • significance: high
  • originality: high
  • clarity: top
  • formatting: perfect
  • grammar: excellent

Author:  Damiano F. G. Fiorillo  on 2022-09-01  [id 2779]

(in reply to Report 1 on 2022-08-30)

We thank the referee for their careful reading and useful comments on our work. We agree with all of them, and we have accordingly changed the text and proceeded to send a new submission. We also attach a pdf version of the revised draft with the changes highlighted in red.

Attachment:

DraftProceedingRed.pdf

Anonymous on 2022-09-01  [id 2785]

(in reply to Damiano F. G. Fiorillo on 2022-09-01 [id 2779])

Thank you.

---

## Editorial Decision

published